# SARS-CoV-2 Infection in Unvaccinated High-Risk Pregnant Women in the Bronx, NY, USA Is Associated with Decreased Apgar Scores and Placental Villous Infarcts

**DOI:** 10.3390/biom13081224

**Published:** 2023-08-06

**Authors:** Sandra E. Reznik, Patricia M. Vuguin, Alexa Cohen, Rasha Khoury, Olivier Loudig, Ridin Balakrishnan, Susan A. Fineberg, Francine Hughes, Malini Harigopal, Maureen J. Charron

**Affiliations:** 1Department of Pathology, Albert Einstein College of Medicine, Bronx, NY 10461, USA; 2Department of Obstetrics and Gynecology and Women’s Health, Albert Einstein College of Medicine, Bronx, NY 10461, USA; 3Department of Pharmaceutical Sciences, St. John’s University, Queens, NY 11439, USA; 4Department of Pediatrics, Columbia University Vagelos College of Physicians & Surgeons, New York, NY 10032, USA; 5Obstetrics and Gynecology and Divisions of Complex Family Planning and Maternal Fetal Medicine, Boston University School of Medicine, Boston, MA 02118, USA; 6Center for Discovery and Innovation, Hackensack Meridian Health, Nutley, NJ 07110, USA; 7Department of Pathology, Louisiana School of Medicine, LSU Health, New Orleans, LA 70112, USA; 8Department of Obstetrics and Gynecology, Massachusetts General Hospital, Boston, MA 02114, USA; 9Department of Pathology, Yale University School of Medicine, New Haven, CT 06519, USA; 10Department of Biochemistry, Albert Einstein College of Medicine, Bronx, NY 10461, USA; 11Department of Medicine and the Fleischer Institute for Diabetes and Metabolism, Albert Einstein College of Medicine, Bronx, NY 10461, USA

**Keywords:** SARS-CoV-2, placental pathology, apgar scores, gestational age, under-resourced patient population

## Abstract

Babies born to severe acute respiratory syndrome corona virus-2 (SARS-CoV-2)-infected mothers are at greater risk for perinatal morbidity and more likely to receive a neurodevelopmental diagnosis in the first year of life. However, the effect of maternal infection on placental function and neonatal outcomes varies depending upon the patient population. We set out to test our hypothesis that maternal SARS-CoV-2 infection in our underserved, socioeconomically disadvantaged, mostly unvaccinated, predominantly African American and Latina population in the Bronx, NY would have effects evident at birth. Under IRB approval, 56 SARS-CoV-2-positive patients infected during the “first wave” of the pandemic with alpha and beta strains of the virus, 48 patients infected during the “second wave” of the pandemic with delta and omicron strains and 61 negative third-trimester high-risk patients were randomly selected from Montefiore Medical Center (MMC), Bronx, NY. In addition, two positive cases from Yale New Haven Hospital, CT were included as controls. All 104 placentas delivered by SARS-CoV-2-positive mothers were uninfected by the virus, based on immunohistochemistry, in situ hybridization, and qPCR analysis. However, placental villous infarcts were significantly increased in first-wave cases compared to second-wave cases or negative controls. Significantly lower Apgar scores at 1 min and 5 min were observed in neonates born to infected mothers with severe symptoms. These findings suggest that even without entering the placenta, SARS-CoV-2 can affect various systemic pathways, culminating in altered placental development and function, which may adversely affect the fetus, especially in a high-risk patient population such as ours. These results underline the importance of vaccination among pregnant women, particularly in low-resource areas.

## 1. Introduction

In line with the well-established link between prenatal exposure to stress and adverse effects on health [1,2], the harmful effects that coronavirus disease 2019 (COVID-19) has on pregnancy, the placenta, the in utero environment, and neonatal outcomes are now coming to light. A recent study of SARS-CoV-2-infected placentas associated with intrauterine fetal demise has established the histopathologic features of SARS-CoV-2 placentitis [3]. Most cases of maternal infection, however, do not lead to infection of the placenta or vertical transmission to the fetus [4,5,6,7]. Nevertheless, maternal SARS-CoV-2 infection even in the absence of infection of the placenta or fetus has been associated with various adverse obstetrical outcomes, including shorter gestational period [8,9,10,11], lower Apgar scores [12], increased rates of intrauterine fetal demise [12], increased rates of neurocognitive deficits in the first year of life [13] and various placental histopathologic findings [14,15,16], depending on the patient population studied.

At the height of the COVID-19 pandemic, New York City was considered the epicenter of the pandemic. Within New York City, the Bronx has been the most affected borough, with over 500,000 cases and more than 8000 deaths to date [17]. Within the Bronx, the Department of Obstetrics and Gynecology and Women’s Health at the Montefiore Medical Center (MMC) serves an under-resourced, predominantly Latina and African American patient population who have increased risk factors for pregnancy complications, including hypertensive disorders of pregnancy and diabetes. We hypothesized that SARS-CoV-2 infection in unvaccinated pregnant minority women in the third trimester would have an unfavorable effect on placental morphology and function, resulting in compromise of the in utero environment and adverse neonatal outcomes when fetal demands for oxygen and nutrients are highest. Babies born to mothers who experience stressors associated with pandemics are at higher risk for developmental programming of cardiometabolic disease and certain cancers [18]. Given the prevalence of infection in pregnant women at the height of the pandemic, a better understanding of how COVID-19 affects the milieu of the developing fetus, which may lead to the identification of histopathological biomarkers for predicting long-term outcomes of exposed offspring, is critical to meet the health needs of the next generation. Here, in cases from two major waves of SARS-CoV-2 infection, we evaluate the placenta of mainly unvaccinated minority patients as well as perinatal clinical outcomes, both of which set the stage for lifelong health and susceptibility to disease.

## 2. Methods

### 2.1. Study Approval

IRB approval was obtained from MMC (Approval Number 2020-11651) and from Yale University Medical Center, New Haven, CT (Yale) (HIC no. 2000027690).

### 2.2. Patient Population

Fifty-six third-trimester placentas from SARS-CoV-2-positive obstetrical patients delivering in 2020 (“first-wave,” alpha/beta cases), 48 third-trimester placentas from SARS-CoV-2-positive patients delivering in 2021–2022 (“second-wave,” delta/omicron cases), and 61 third-trimester placentas from SARS-CoV-2-negative patients were randomly selected from the MMC pathology archives with approval from the Albert Einstein College of Medicine Institutional Review Board (IRB). Patients with diabetes mellitus (gestational, type 1 or type 2 diabetes) were excluded from the study. Patients in all three groups were matched for maternal age and body mass index (BMI), as shown in Table 1. In addition, one previously described [8] SARS-CoV-2-positive mid-trimester placenta and one newly identified SARS-CoV-2-positive third trimester from Yale New Haven Hospital, New Haven, CT were included, with approval from the Yale IRB. All first-wave, alpha/beta patients were unvaccinated for SARS CoV-2. Nine of the 48 second-wave, delta/omicron patients had received at least one SARS-CoV-2 vaccine at the time of delivery.

### 2.3. Histopathologic Evaluation

Placentas from mothers infected with the first and second waves of SARS-CoV-2 and the randomly selected third-trimester placentas delivered by uninfected mothers were compared histopathologically, as detailed in Table 2. Four Hematoxylin and Eosin (H and E) stained sections of formalin-fixed paraffin-embedded (FFPE) tissue from each placenta from the MMC pathology archives (n = 56 from first-wave SARS-CoV-2-positive patients, n = 48 from second-wave SARS-CoV-2-positive patients and n = 61 from SARS-CoV-2-negative patients) were examined by a practicing pathologist at MMC, who was blinded as to SARS-CoV-2 infection status, using routine light microscopy. Sections included at least one extra-embryonic membrane roll, two sections of umbilical cord, at least one section of villous parenchyma including the chorionic plate and at least two sections of villous parenchyma including the decidua basalis. All placentas were evaluated for several features of maternal vascular malperfusion (distal villous hypoplasia, villous congestion, villous infarct, intervillous thrombi and decidual vasculopathy in the decidua capsularis and decidua basalis). In addition, all cases were examined for inflammatory responses, including chorioamnionitis and funisitis. No cases showed evidence of fetal vascular malperfusion, chronic villitis or chronic histiocytic intervillositis. The criteria used for making the various histopathologic diagnoses are provided in Appendix A.

### 2.4. Quantitative Polymerase Chain Reaction (qPCR)

#### 2.4.1. FFPE Tissue Acquisition

Unstained 10 μm sections from one FFPE tissue block from each of the 56 MMC SARS-CoV-2-positive study subjects from the first wave and SARS-CoV-2-infected lung tissue as a positive control were obtained. The corresponding H and E-stained sections were reviewed by one of the study pathologists (SER) for uniform representation of the placental cells on the sections. Unstained tissue sections in their entirety were subjected to RNA extractions, prior to qPCR analyses, using the Taqman^®^ (Thermo Fisher Scientific Inc., Waltham, MA, USA) 2019-nCov Assay Kit.

#### 2.4.2. FFPE RNA Extraction and Quantification

Total RNA extractions from FFPE clinical specimens were performed using a simultaneous RNA/DNA extraction protocol described previously, where RNA is extracted prior to DNA [19,20]. In brief, a total of 3 consecutive unstained 10 µm tissue sections were scraped off the slides and transferred to individual siliconized Eppendorf tubes. The tissues were de-paraffinized using Citri-Solv (Thermo Fisher Scientific Inc.) at room temperature on a Thermomixer (Eppendorf, Enfield, CT, USA), followed by ethanol washes on ice, one 1xPBS (RNase-free, Thermo Fisher Scientific Inc.) wash, and a re-hydration in the presence of RNase-inhibitors in 1xPBS (RNase-free). The tissues were subjected to proteinase K (3 mg/mL) digest at 59 °C for 1 h. The digested tissues underwent butanol-1 extraction to obtain a final volume of 100 μL, which was homogenized in 1 mL of TRIzol (Invitrogen, Carlsbad, CA, USA) following the manufacturer’s instructions. The RNA was recovered from the upper phase of the TRIzol solution, transferred to a new siliconized Eppendorf and precipitated with 0.1 μL/μL linear acrylamide (1 μL), 3M sodium acetate (18 μL), and 660 μL of isopropanol. The tubes were stored at −20 °C overnight and centrifuged the next day at 14,000 rpm for 30 min at 4 °C. The RNA pellets were washed with 200 μL of 70% RNase-free ethanol, dried and re-suspended in 12 μL of RNase-free 1xTris/EDTA solution and incubated for 30 min at 70 °C. The RNA was quantified on a total RNA chip on a Bioanalyzer (Agilent Technologies, Wilmington, DE, USA). Genomic DNA was extracted from the lower phase of TRIzol and stored for future experiments [21].

#### 2.4.3. SARS-CoV-2 Reverse Transcription and qPCR Analyses

Quantitative PCR experiments were performed using Taqman^®^ 2019-nCov Assay Kit v1, Taqman^®^ 2019 nCov Control Kit v1 and Taqman^®^ Fast Virus 1-Step Master Mix (Applied Biosystems Inc., Norwalk, CT, USA). For each clinical specimen, 100 ng of total RNA was used for each qPCR analysis. Reagents for each RNA specimen were combined in a MicroAmp Fast Optical 96-well plate following the manufacturer’s instructions. In brief, 6.25 μL of Taqman^®^ Fast Virus 1-Step Master Mix (4x), 1.25 μL nCov assay (20x) (either ORF1ab, S Protein, or N Protein), 1.25 μL of RNaseP assay (20x) and 11.25 μL of nuclease-free water were combined in each well. For each RNA sample, 5 μL of total RNA (20 ng/mL) were added to respective sample wells of a barcoded 96-well plate (Applied Biosystems Inc.). For these analyses, following the manufacturer’s recommendations, control wells containing 1 μL of either ORF1ab, S Protein, or N Protein with 4 μL of nuclease-free water were also prepared for each 96-well plate. The real-time PCR quantifications for each sample were monitored on a StepOnePlus instrument (Applied Biosystems Inc.), and the comparative thresholds were determined after sample plates underwent the recommended cycles as follows: 5 min at 50 °C, 20 s at 95 °C, 3 s at 95 °C, and 30 s at 60 °C for 40 cycles. The fold-change differences between the clinical specimens and the control specimens were calculated using the 2^(ΔΔCT)^ formula.

### 2.5. Immunohistochemistry (IHC)

FFPE 4 μm sections from all 104 MMC SARS-CoV-2-positive study subjects (56 first-wave subjects and 48 second-wave subjects) as well as the two positive control Yale cases were stained with anti-SARS-CoV-2 nucleocapsid protein antibody (Thermo Fisher Scientific Inc.), mouse monoclonal antibody clone B46F, dilution 1:200), as previously described [8]. The nucleocapsid antibody was previously validated with positive and negative controls. The antibody immunostains SARS-CoV-2-infected skin, lung, liver and kidney autopsy tissue [22]. In addition, the antibody used in this study immunostains control infected placental tissue [8], which contains SARS-CoV-2 RNA detectable by two different spike protein antibodies (Sino Biological Inc., Chesterbrook, PA, USA, 40150-T62-COV; and GeneTex, Irvine, CA, SARS-CoV/SARS-CoV-2 spike antibody clone 1A9), by PCR and by ISH [8]. The nucleocapsid antibody has also been shown not to stain placental tissue that is negative for SARS-CoV-2 by PCR and by ISH and does not cross-react with human coronaviruses 229E or OC43 [22].

### 2.6. In Situ Hybridization (ISH)

ISH was performed on sections from the 56 MMC SARS-CoV-2-positive study subjects and the two positive control cases from Yale with a probe directed against the spike protein (V-nCoV2019-S; Advanced Cell Diagnostics, Hayward, CA, USA), using appropriate positive and negative controls [21]. Positive control placental tissue was proven to be infected with SARS-CoV-2 by PCR and by immunohistochemistry, using two different antibodies directed against the spike protein (Sino Biological Inc., 40150-T62-COV; and GeneTex, SARS-CoV/SARS-CoV-2 spike antibody clone 1A9) and one antibody directed against nucleocapsid protein (Thermo Fisher Scientific Inc., mouse monoclonal antibody clone B46F) [22]. Negative controls were proven to be uninfected, based on PCR and IHC, using the antibodies listed. Probes directed against the bacterial gene dapB and probes for the housekeeping gene peptidylprolyl isomerase B were included as negative and positive technical controls, respectively. Sections were counterstained with periodic acid-Schiff.

### 2.7. Statistical Analysis

Statistical analyses were performed using JMP software version 7.0 (SAS Institute, Cary, NC, USA) or Graph Pad Prism software version 5.04 for Windows (Graph Pad Software, San Diego, CA, USA). For continuous variables, data are presented as the means ± SEM. Analysis of variance was used to test for significant differences between the means of two (*t*-test) or more groups, and Bonferroni’s post hoc analysis was performed. Pearson’s χ^2^ test was the statistical test applied to sets of categorical data to evaluate how likely it is that any observed difference between the sets arose by chance. When appropriate, the two-tailed *U*-test was used. Acceptable study power was agreed a priori to be ≥ 80% (type I error of ≤ 0.20). *p* < 0.05 was considered statistically significant.

## 3. Results

### 3.1. Severe Maternal SARS-CoV-2 Infection Affects Birth Weight and Apgar Scores

Fifty-six first-wave SARS-CoV-2-positive, 48 second-wave SARS-CoV-2-positive and 61 SARS-CoV-2-negative MMC third-trimester obstetrical patients were compared for sociodemographic, obstetrical, and medical parameters, as detailed in Table 1. None of the neonates developed clinical, radiologic, hematologic, or biochemical evidence of COVID-19. While patients infected during the first wave were more often asymptomatic than patients infected during the second wave, among symptomatic women, patients infected in the second wave presented with mild symptoms much more commonly (*p* < 0.01). Although all three groups had a high prevalence of obesity, no significant differences among the three groups were found in age, body mass index (BMI), cigarette smoking, gravidity, parity, or mode of delivery. Interestingly, while there were no significant differences in race between the negative control group and the first-wave group, the second-wave group comprised significantly fewer Latina patients than the control and first-wave groups (*p* < 0.00001, *p* < 0.00001) and significantly more non-Latina white patients than the control and first-wave groups (*p* < 0.05, *p* < 0.001) (Table 1).

In Table 3, all the SARS-CoV-2-positive patients for whom data related to symptoms were available (both first- and second-wave patients, n = 95) were classified as having no symptoms (n = 57), mild symptoms (n = 28), or severe symptoms (n = 10). Neonatal outcomes, including birth weights, Apgar scores at 1 min and Apgar scores at 5 min were then compared by ANOVA. Of the ten cases with severe symptoms, six were from the first wave (alpha/beta strains) and four were from the second wave, with two delta-strain and two omicron-strain cases. The presence of severe symptoms was significantly linked to decreased Apgar scores at both 1 min (*p* < 0.05) and 5 min (*p* < 0.05). While the study was most likely not adequately powered to detect a significant effect on birth weight, a downward trend was observed in babies born to mothers with severe symptoms (*p* = 0.06) (Table 3).

### 3.2. First-Wave Maternal SARS-CoV-2 Infection Increases Placental Villous Infarcts

The prevalence of placental histopathology was very high in the cases examined, not surprisingly, given the high frequencies of placental findings in our high-risk patient population. Nevertheless, a statistically significant difference between the placentas from first-wave patients and negative controls was seen in the number of villous infarcts with placentas from the infected mothers being more frequently infarcted (*p* < 0.05, Table 2). Examples of villous infarcts observed in placentas delivered by first-wave infected mothers are shown in Figure 1A–D.

### 3.3. The Placenta Is an Effective Barrier for SARS-CoV-2

Three methods were used to test for the presence of SARS-CoV-2 in the placentas of the positive MMC patients: qPCR; ISH, using probes directed against SARS-CoV-2; and IHC, using an antibody directed against the nucleocapsid protein. All 104 cases were subjected to IHC, all 56 first-wave cases were tested by qPCR and a subset of 19 first-wave cases were subjected to ISH. All 104 cases were consistently negative for SARS-CoV-2. This finding is consistent with the low rate of placental infection reported by other investigators [4,5,6,7], as well as data suggesting that ACE2 and transmembrane protease serine 2 (TMPRSS2) rarely co-localize in the placenta [23].

However, two control infected placentas clearly showed positive staining in the syncytiotrophoblast, using IHC (Figure 2A and Figure 2C, respectively) and ISH (Figure 2B and Figure 2D, respectively). The mid-trimester placenta used as a positive control (Figure 2A,B) was delivered by dilation and evacuation at 22 weeks’ gestation by a 35-year-old gravida 3 para 1011 patient with severe preeclampsia complicated by placental abruption and disseminated intravascular coagulation. Microscopic findings in this placenta were significant for a marginal retroplacental hematoma and overlying villous infarct (consistent with placental abruption) and for diffuse intervillous fibrin and CD68- and CD3-positive mononuclear cell infiltrates, consistent with macrophages and lymphocytes, respectively, and with SARS-CoV-2 placentitis. Electron microscopy of this placenta revealed cytoplasmic viral particles in the villous trophoblast cells [8]. The third-trimester placenta used as a second positive control (Figure 2C,D) was delivered at 34 weeks and 6 days’ gestation by a 40-year-old gravida 6 para 3103 mother with no significant obstetrical history other than preterm labor. She was treated prophylactically with 17-alpha-hydroxyprogesterone (Makena). Microscopically, this placenta was significant for increased intervillous fibrin and multiple acute villous infarcts. In addition, like the first positive control case, there was CD68 positive histiocytic intervillositis, consistent with SARS-CoV-2 placentitis. SARS-CoV-2 was confirmed in both positive control patients by PCR of nasopharyngeal swabs.

## 4. Discussion

While the effect of maternal SARS-CoV-2 infection on pregnancy has been investigated in various patient cohorts [9,12,13], we evaluate here the effect of maternal infection in an under-resourced, predominantly African American and Latina, high-risk patient population. In addition, we compare clinical outcomes and placental histopathology in patients infected with alpha and beta strains of SARS-CoV-2 (first-wave cases) and delta and omicron strains of SARS-CoV-2 (second-wave cases). While the patients are well matched for maternal age, BMI, gravidity, parity and mode of delivery across all groups, surprisingly, none of the patients infected in the second wave was Latina. Although it would be of interest to test whether vaccination status affected either clinical outcome or placental pathology, among our 104 infected patients, only nine were vaccinated; hence, the study was not adequately powered for effects of vaccination status to be observed. However, other studies support the importance of SARS-CoV-2 vaccination of pregnant women, specifically the advantages of vaccination during the second and third trimesters [24]. In fact, Schwartz et al. have reported a correlation between vaccination status and placental pathology [25].

While placental SARS-CoV-2 infection is rare, COVID-19 pneumonia or infection of other non-gynecologic organs was common in pregnant women in our center with 30% of patients presenting to our Labor and Delivery service testing positive for SARS-CoV-2 by nasopharyngeal swab PCR at the height of the first wave of the pandemic. Our infection rate was higher than those in nearby institutions in greater New York serving different patient populations, which averaged only 14% infection rates [26,27,28]. Both the rate of maternal infection and the impact of maternal COVID-19 on offspring is likely to be more pronounced in patient populations like ours, where underlying risk factors for obesity, hypertension and diabetes are common. The results of this study underline the importance of vaccination among pregnant women, particularly in low-resource areas.

One of the significant findings in our study is that maternal SARS-CoV-2 infection is associated with Apgar scores that are approximately two points lower at 1 and 5 min in patients with severe symptoms. Decreases in Apgar scores of this magnitude predict greater rates of neonatal morbidity and mortality [29]. Interestingly, in a recent study that examined a more affluent, predominantly Caucasian patient population in Boston, MA, investigators found no significant change in Apgar scores associated with maternal SARS-CoV-2 infection [13], suggesting that social determinants of health may exacerbate health risks for offspring of infected pregnant women. Importantly, the SARS-CoV-2-positive patients in this study were infected at the time of delivery and most likely not infected in the first weeks of gestation, when placentation and critical uteroplacental vascular remodeling is taking place. The effect of COVID-19 and loss of ACE2 in early pregnancy on placental morphology and function and fetal wellbeing remains to be determined. Of note, Resta et al. have recently reported statistically significant correlations between symptoms of SARS-CoV-2 viremia and placental histopathology, decidual arteriopathy, thrombosis and loss of angiotensin-converting enzyme-2 expression [30]. It is unclear why we only observed an effect on placental histopathology in first-wave cases, given that our second-wave cases included the delta variant of SAR-CoV-2, the most virulent strain observed to date. The different viral strains may have different mechanisms of placental injury independent of their systemic virulence.

One weakness of our study is the fact that our second-wave cases include infections caused by two very different strains of SARS-CoV-2. Although delta and omicron strains of the virus emerged in the later stages of the pandemic, when treatment protocols to combat infection were better developed, the delta strain proved to be much more virulent than the omicron strain. Larger studies are needed to tease out how these two strains of SARS-CoV-2 differ in their effects in pregnancy.

Another weakness of our study is the very high background of placental pathology in our control cases. The prevalence of placental pathology in our cases is typical of our high-risk patient population comprising predominantly Latina and African American women, who have increased risk factors for obstetrical complications. Moreover, because maternal SARS-CoV-2 infection per se is a criterion for placental examination, theoretically, placentas from infected mothers in this study might include ones with or without pathology. On the other hand, placentas delivered by uninfected mothers are only examined if there is a risk factor for placental abnormality. Hence, the study is hedged in favor of finding pathology in uninfected cases. Nevertheless, even in the context of a high background level of placental histopathology, placental villous infarcts were significantly increased in placentas delivered by mothers infected in the first SARS-CoV-2 wave. Villous infarcts represent the terminal lesion affecting the placental parenchyma in the setting of maternal vascular malperfusion. Infarcted villous parenchyma is no longer able to carry out gas exchange, nutrient delivery and waste removal between the mother and fetus. While there is “placental reserve,” the presence of infarcts also serves as a biomarker for poor placental perfusion and function. Poor placental function increases the risk for adverse perinatal outcomes, including intrauterine growth restriction and fetal demise. In addition, suboptimal in utero environments, such as ones in which the placenta is insufficient, result in altered fetal epigenetic programming and lifelong increased risk for cardiometabolic disease [31].

A strength of the study is that the patients in our three groups were well matched for several clinical parameters, including BMI and rate of Cesarean delivery. Moreover, patients with diabetes were excluded from the study. On the other hand, hypertensive disorders of pregnancy were not included as one of the potential variables.

Although the neonates in the SARS-CoV-2-positive groups were not infected, maternal SARS-CoV-2 infection altered their in utero environment. In addition to potential effects on the fetal milieu produced by the maternal immune response to the virus, changes in levels of ACE2, the SARS-CoV-2 receptor, are directly linked to maternal vascular malperfusion of the placenta and obstetrical complications [32,33,34,35,36]. The long-term consequences of these complications will become apparent in the decades ahead but based on our current understanding of the effect of the in utero environment on lifelong fetal programming, it is likely that offspring born to SARS-CoV-2-positive mothers will be at increased risk for cardiometabolic disease, particularly among high-risk patient populations such as ours.

## Figures and Tables

**Figure 1 biomolecules-13-01224-f001:**
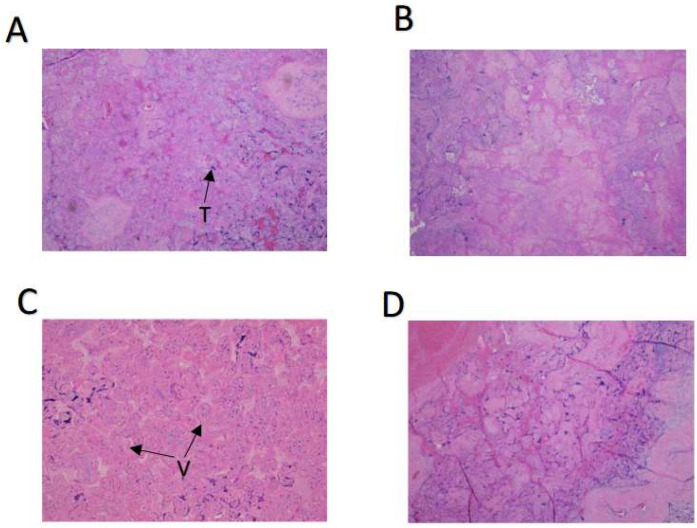
Placental villous infarcts associated with maternal SARS-CoV-2 infection. Discrete areas of coagulative necrosis, comprising ghost villi and faded syncytiotrophoblasts, compatible with subacute to chronic villous infarcts, in placentas from four infected women. Microscopic slides were originally photographed at a magnification of 200×. SARS-CoV-2, severe acute respiratory syndrome coronavirus 2; T, trophoblast; V, chorionic villi.

**Figure 2 biomolecules-13-01224-f002:**
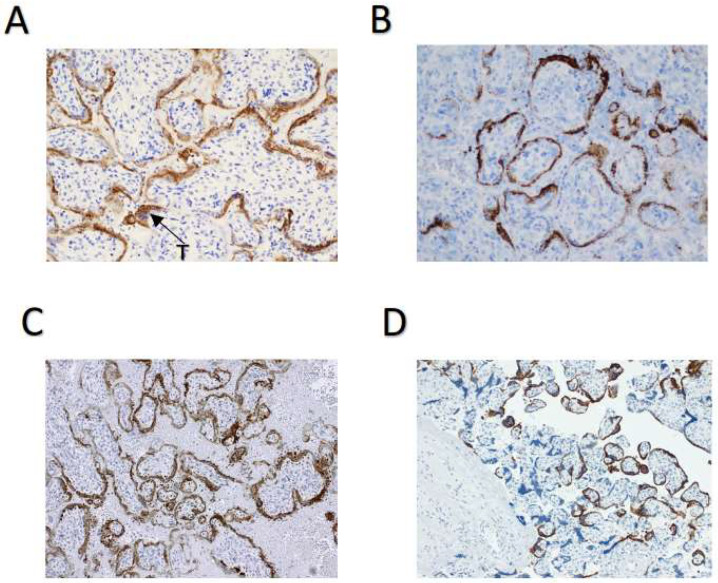
Positive control cases of SARS-CoV-2 placentitis. (**A**) SARS-CoV-2 IHC staining in a 22-week infected placenta. Sections were stained using an antibody directed against the nucleocapsid protein. (**B**) SARS-CoV-2 ISH staining of the same placenta as in Panel A. Hybridization was directed against the Spike protein. (**C**) SARS-CoV-2 IHC staining in a 34-week-and-6-day infected placenta. (**D**) SARS-CoV-2 ISH in the same placenta as in Panel C. Microscopic slides were originally photographed at a magnification of 200×. SARS-CoV-2, severe acute respiratory syndrome coronavirus 2.

**Table 1 biomolecules-13-01224-t001:** Clinical features of patients who were uninfected, infected in the first wave (alpha/beta) and infected in the second wave (delta/omicron) of the SARS CoV-2 pandemic. * *p* < 0.05, ** *p* < 0.01, **** *p* < 0.0001, ***** *p* < 0.00001.

	SARS-CoV-2-Negative (n = 61)	SARS-CoV-2-Positive, First Wave (n = 56)	SARS-CoV-2-Positive, Second Wave (n = 48)	*p*, First Wave vs. Negative	*p*, Second Wave vs. Negative	*p*, First Wave vs. Second Wave
Severity of Infection (%)						
--Asymptomatic	N/A	63.2	47.8	N/A	N/A	NS
--Mild	N/A	19.3	39.1 **	N/A	N/A	*p* < 0.01
--Severe	N/A	17.5	13.1	N/A	N/A	NS
Mean age ± SEM (Years)	30.92 ± 1.20	29.42 ± 1.08	30.03 ± 1.57	NS	NS	NS
Mean BMI ± SEM (kg m^−2^)	32.76 ± 1.76	32.00 ± 1.46	31.61 ± 1.78	NS	NS	NS
Race (%)						
--Black	52	47	43	NS	NS	NS
--Latina	40	42	0 *****	NS	*p* < 0.00001	*p* < 0.00001
--Non-Latina white	4.0	5.3	35 *	NS	*p* < 0.05	*p* < 0.0001
--Asian	4.0	5.3	11	NS	NS	NS
Cigarette smoking (%)	12	0	4.3	NS	NS	NS
Mean Gravidity ± SEM	2.9 ± 0.2	3.0 ± 0.2	3.3 ± 0.5	NS	NS	NS
Mean Parity ± SEM	1.2 ± 0.2	1.8 ± 0.2	1.4 ± 0.4	NS	NS	NS
Mode of delivery (%)						
--Cesarean	39.3	38.6	39.2	NS	NS	NS
--Vaginal	60.7	61.4	60.8	NS	NS	NS
Mean Apgar scores ± SEM						
--Apgar score at 1 min	8.21 ± 0.23	7.56 ± 0.37	7.42 ± 0.73	NS	NS	NS
--Apgar score at 5 min	8.63 ± 0.21	7.94 ± 0.36	7.95 ± 0.65	NS	NS	NS
Mean gestational age ± SEM (weeks)	37.5 ± 0.7	37.2 ± 0.7	38.2 ± 0.5	NS	NS	NS
Mean Birthweight ± SEM (g)	3079 ± 88	3046 ± 91	2951 ± 190	NS	NS	NS
Fetal death (%)	12	0	5	NS	NS	NS

**Table 2 biomolecules-13-01224-t002:** Histopathologic features of placentas from patients who were uninfected, infected in the first wave (alpha/beta) and infected in the second wave (delta/omicron) of the SARS CoV-2 pandemic.

	SARS CoV-2-Negative (n = 61)	SARS-CoV-2-Positive, First Wave (n = 56)	SARS-CoV-2-Positive, Second Wave (n = 48)	*p*, First Wave vs. Negative	*p*, Second Wave vs. Negative	*p*, First Wave vs. Second Wave
Maternal vascular malperfusion (%)						
--Small for gestational age placenta	23.0	12.5	37.5	NS	NS	NS
--Distal villous hypoplasia	8.2	7.1	6.3	NS	NS	NS
--Villous congestion	8.2	5.4	12.5	NS	NS	NS
--Villous infarct	6.6	19.6	10.4	*p* < 0.05	NS	NS
--Intervillous thrombus	11.5	10.7	10.4	NS	NS	NS
--Decidual vasculopathy	1.7	1.8	0	NS	NS	NS
Inflammatory responses (%)						
----Chorioamnionitis	34.4	28.6	39.6	NS	NS	NS
----Funisitis	16.7	10.7	14.6	NS	NS	NS

**Table 3 biomolecules-13-01224-t003:** Birth weights and Apgar scores of babies born to patients who were uninfected, infected with mild symptoms and infected with severe symptoms of SARS-CoV-2.

	SARS-CoV-2-Negative (n = 61)	SARS-CoV-2-Positive Asymptomatic (n = 57)	SARS-CoV-2-Positive Mild Symptoms (n = 28)	SARS-CoV-2-Positive Severe Symptoms (n = 10)	*p*
Mean birth weight ± SEM (g)	3079 ± 88	3000 ± 92	3210 ± 135	2595 ± 191	*p* = 0.06
Mean Apgar scores ± SEM					
--Apgar score at 1 min	8.21 ± 0.31	7.78 ± 0.32	7.85 ± 0.47	6.06 ± 0.63	*p* < 0.05
--Apgar score at 5 min	8.63 ± 0.29	7.85 ± 0.46	8.07 ± 0.43	6.66 ± 0.59	*p* < 0.05

## Data Availability

Not applicable.

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
