# Peer review of "SARS-CoV-2 Infection in Unvaccinated High-Risk Pregnant Women in the Bronx, NY, USA Is Associated with Decreased Apgar Scores and Placental Villous Infarcts"

_biomolecules, 2023, doi:10.3390/biom13081224_

Round 1
Reviewer 1 Report (Previous Reviewer 1)
Now You have separated your cases into “first wave” and “second wave” but in reality the omicron wave with the most asymptomatic patients should be considered separately from delta wave with more symptomatic cases; however, this limitation should be underlined otherwise you risk giving a false message that delta wave is less severe and similar to omicron. Anyway you must at least specify which wave the SARS-CoV-2 Positive patients with Severe Symptoms (N.10) belong to.
It is useful to add in Discussion on line 252 that other studies have recently described the possible differences between vaccinated and non-vaccinated patients, demonstrating that vaccination during the second and third trimester of pregnancy against COVID-19 is strongly recommended. Enter about it the reference“Vimercati A, R De Nola, S Battaglia , R Di Mussi, G Cazzato , L Resta, M Chironna, D Loconsole, L Vinci, G Chiarello, M Marucci and E Cicinelli . Adverse Maternal Outcomes in Pregnant Women Affected by Severe-Critical COVID-19 Illness: Correlation with Vaccination Status in the Time of Different Viral Strains’ Dominancy . Vaccines 2022, 10, 2061, doi: 10.3390/vaccines10122061. “
In discussion line 307-results without s
On line 296 in Discussion you can add another important reference of a study showing that the symptoms (likely to be related to viremia) were statistically significantly correlated with placental histopathological changes, such as maternal malperfusion, decidual arteriopathy, blood vessel thrombus and loss of ACE- “ Resta L, Vimercati A, Cazzato G, Fanelli M, Scarcella SV, Ingravallo G, Colagrande A, Sablone S, Stolfa M, Arezzo F, Lettini T, Rossi R. SARS-CoV-2, Placental Histopathology, Gravity of Infection and Immunopathology: Is There an Association? Viruses. 2022 Jun 18;14(6):1330. doi: 10.3390/v14061330
Author Response
Now You have separated your cases into “first wave” and “second wave” but in reality the omicron wave with the most asymptomatic patients should be considered separately from delta wave with more symptomatic cases; however, this limitation should be underlined otherwise you risk giving a false message that delta wave is less severe and similar to omicron. Anyway you must at least specify which wave the SARS-CoV-2 Positive patients with Severe Symptoms (N.10) belong to.
We thank Reviewer 1 for this helpful comment. We have now underlined this limitation. We have also provided the breakdown of the severe cases among the different viral strains.
It is useful to add in Discussion on line 252 that other studies have recently described the possible differences between vaccinated and non-vaccinated patients, demonstrating that vaccination during the second and third trimester of pregnancy against COVID-19 is strongly recommended. Enter about it the reference“Vimercati A, R De Nola, S Battaglia , R Di Mussi, G Cazzato , L Resta, M Chironna, D Loconsole, L Vinci, G Chiarello, M Marucci and E Cicinelli . Adverse Maternal Outcomes in Pregnant Women Affected by Severe-Critical COVID-19 Illness: Correlation with Vaccination Status in the Time of Different Viral Strains’ Dominancy . Vaccines 2022, 10, 2061, doi: 10.3390/vaccines10122061. “
We have added this reference.
In discussion line 307-results without s
We have made this correction.
On line 296 in Discussion you can add another important reference of a study showing that the symptoms (likely to be related to viremia) were statistically significantly correlated with placental histopathological changes, such as maternal malperfusion, decidual arteriopathy, blood vessel thrombus and loss of ACE- “ Resta L, Vimercati A, Cazzato G, Fanelli M, Scarcella SV, Ingravallo G, Colagrande A, Sablone S, Stolfa M, Arezzo F, Lettini T, Rossi R. SARS-CoV-2, Placental Histopathology, Gravity of Infection and Immunopathology: Is There an Association? Viruses. 2022 Jun 18;14(6):1330. doi: 10.3390/v14061330
We have added this reference.
Reviewer 2 Report (Previous Reviewer 2)
The authors provided responses to reviwer's commnets. It would be of interest showing the lesions that had a statistically different distribution among the groups, e.g placental infarcts
Author Response
We thank Reviewer 2 for this suggestion. We have added a figure illustrating the placental villous infarcts.
Reviewer 3 Report (New Reviewer)
The study design appears appropriate, and the methods used for placental evaluation and testing for SARS-CoV-2 infection are well-described. The clear presentation of results and statistical analyses enhances the clarity and validity of the findings.
One point to consider is the potential confounding effect of other risk factors such as obesity, hypertension, and diabetes, which are prevalent in the study population. It might be helpful to discuss how the researchers accounted for or controlled for these confounding factors in their analysis.
One limitation of the study is the high background level of placental pathology observed in control cases. Further discussion on how this may have influenced the study's findings and the potential confounding effects is important.
Author Response
One point to consider is the potential confounding effect of other risk factors such as obesity, hypertension, and diabetes, which are prevalent in the study population. It might be helpful to discuss how the researchers accounted for or controlled for these confounding factors in their analysis.
We now discuss these potential confounding variables and to what extent we were able to control for them.
One limitation of the study is the high background level of placental pathology observed in control cases. Further discussion on how this may have influenced the study's findings and the potential confounding effects is important.
We have extended our discussion about this limitation and added the interesting point that due to the fact that all placentas from infected patients were automatically examined, the study is actually hedged in favor of finding more pathology in uninfected cases. It is therefore even more significant that we found increased pathology in the infected cases.
Reviewer 4 Report (New Reviewer)
The authors have designed an original and interesting retrospective investigation that compares the demographic & clinical features of 2 cohorts of pregnant women infected with SARS-CoV-2 during different temporal phases of the COVID-19 pandemic (Phase I alpha & beta variants) and Phase II (delta and omicron) and correlates these data with perinatal outcomes and placental pathology from a hospital population of socially and financially disadvantaged women from Bronx, New York. The overall nature of this study is by nature complex involving multiple risk co-variables and potentially confounding behavioral, obstetrical and other medical factors. However, the authors are to be complemented as performing an investigation that demonstrates potentially important findings relating to SARS-CoV-2 in pregnancy and its effect on the placenta. The authors performed immunohistochemical and nucleic acid testing methods to determine placental infection among the mothers with COVID-19 and found no evidence of placental infection with SARS-CoV-2. Interestingly, placental infarcts were increased compared with controls during the first wave of COVID-19. The authors discuss this finding as well as their data from analyses of Apgar scores in which scoring was less than among newborns from severely symptomatic infected mothers, and additional co-variables in light of the knowledge of COVID-19 in pregnancy.
This is a well-organized manuscript on a complicated topic. The references appear current, the tables are well-designed, easily read and highlight the text, and the photomicrographs are clear and highly illustrative.
The authors may wish to consider the following suggestions and recommendations in revising their manuscript.
-Lines 90-91. This reviewer is curious as to the rationale of grouping infections with the delta and omicron variants for analysis. The delta wave resulted in some of the worst obstetrical, placental & perinatal outcomes among unvaccinated persons of the pandemic, while the omicron variants have remained relatively benign...
-Line 95-110. Can the authors provide the placental pathology criteria that they used for their diagnoses?
-Line 299-300. Do the authors have a potential theory as to why infects were increased in placentas during the first wave, but not the second wave that included the delta variant? The delta variant produced the most severe effects during pregnancy of all variants.
- Lines 283-284. A 2023 article in the Am J Ob Gyn has addressed the relationship of maternal COVID-19 vaccination and the development of placental pathology. I suggest that you consider citing it... Schwartz DA, Mulkey SB, Roberts DJ. SARS-CoV-2 placentitis, stillbirth, and maternal COVID-19 vaccination: clinical-pathologic correlations. Am J Obstet Gynecol. 2023;228(3):261-269. doi:10.1016/j.ajog.2022.10.001
-Line 334. The references are not properly formatted for the journal and should be revised.
Author Response
-Lines 90-91. This reviewer is curious as to the rationale of grouping infections with the delta and omicron variants for analysis. The delta wave resulted in some of the worst obstetrical, placental & perinatal outcomes among unvaccinated persons of the pandemic, while the omicron variants have remained relatively benign...
A similar point was raised by Reviewer 1 and we thank Reviewer 4 for the query as well. We now include a discussion of how our cases were divided into groups and we mention that a larger study is needed to tease out the differences in effects of the delta and omicron strains.
-Line 95-110. Can the authors provide the placental pathology criteria that they used for their diagnoses?
We have added a table (Supplemental Table 1), which details the placental pathology criteria for our histologic diagnoses.
-Line 299-300. Do the authors have a potential theory as to why infects were increased in placentas during the first wave, but not the second wave that included the delta variant? The delta variant produced the most severe effects during pregnancy of all variants.
We agree that it is curious that we found an increase in pathology in the first wave but not the second wave, given the inclusion of the delta variant in the second wave. We discuss this curiosity in the manuscript. We feel that the mechanisms of placental injury of the different viral strains are not completely understood.
- Lines 283-284. A 2023 article in the Am J Ob Gyn has addressed the relationship of maternal COVID-19 vaccination and the development of placental pathology. I suggest that you consider citing it... Schwartz DA, Mulkey SB, Roberts DJ. SARS-CoV-2 placentitis, stillbirth, and maternal COVID-19 vaccination: clinical-pathologic correlations. Am J Obstet Gynecol. 2023;228(3):261-269. doi:10.1016/j.ajog.2022.10.001
We now include this important reference in the manuscript.
-Line 334. The references are not properly formatted for the journal and should be revised.
We have re-formatted the references to suit the style of this journal.
This manuscript is a resubmission of an earlier submission. The following is a list of the peer review reports and author responses from that submission.
Round 1
Reviewer 1 Report
I think that your hypothesis that maternal SARS-CoV-2 infection in High Risk pregnant women would affect placental histopathology, especially villous infarct, and that the rate of placental infection is low is already well known in the literature
This manuscript could be more interesting if it reported new data such as:
· Divide the results according to the different pandemic waves and, if possible, viral variant. It is known in the literature that over time the alternation of different viral variants has given different results up to the less severe omicron variant.
· The importance of vaccination for SARS-CoV-2 among pregnant women, particularly in low resource areas, would be better stressed comparing the results between Covid 19 vaccinated and unvaccinated pregnant women in the same period
· The low rate of placental infection as well as data suggesting the shorter length of gestation and lower Apgar scores are just known in literature
· On line 12 page of results-you could add to bibliographic items 8-10 a recent article on the same topic with statistical consistentency “Covid-19 Infection in Pregnancy: Obstetrical Risk Factors and Neonatal Outcomes-A Monocentric, Single-Cohort Study.
Reviewer 2 Report
Histopathological review was done by a single pathologist. No reference of blinding from SARS-CoV-2 status is stated and it can not be assumed. Review had to be done blinded to SARS-CoV-2 maternal status.
According to material and methods, the analysys was done upon a cohort of 55+61 placentas. However, Table 1 and Table 2 (displaying clinical features and histolgy respectively) show heads of 25 and 19. Are this figures tha actual number of analyzed cases? What about the information about the other 72 cases? The reader must assume that figures in the tables are percentages (5.3 cannot be an absolute frequancy) but this is not stated.
How do the authors explain that a difference in gestational length between positive and negative mothers is found, whereas according to table 1 there is no such a difference? Also, mild/unspecified preeclampsia is reported as significantly different in the text but non-significant in Table 1. This is inconsistent.
To address if maternal viral infection conferes worse obstetrical or perinatal outcomes the variables known to exert an effect shoud be controlled, eg, look only to patients with the same degree of preeclampsia and report existing differences